# Over-Inflating a Tracheostomy Tube Cuff for Tracheo-Innominate Artery Fistula

**DOI:** 10.3390/diagnostics14020223

**Published:** 2024-01-20

**Authors:** Yi-Chia Hsieh, Wen-Ruei Tang, Ying-Yuan Chen

**Affiliations:** 1Department of Urology, National Cheng Kung University Hospital, College of Medicine, National Cheng Kung University, Tainan 704302, Taiwan; i54006160@gs.ncku.edu.tw; 2Division of Thoracic Surgery, Department of Surgery, National Cheng Kung University Hospital, College of Medicine, National Cheng Kung University, Tainan 704302, Taiwan; tony82217@gmail.com

**Keywords:** tracheo-innominate artery fistula, complication of tracheostomy, computed tomography angiography

## Abstract

We report an angiographic image of a 58-year-old woman with profuse bleeding from a tracheo-innominate artery fistula. It may not have been possible to obtain this valuable image if adequate initial resuscitation and an over-inflated tracheostomy tube cuff had not been administered to stop bleeding during an emergency.

A 58-year-old woman presented to our hospital with profuse bleeding from a deflated cuff tracheal stoma. She had stage IVB anaplastic thyroid carcinoma with upper airway obstruction and had undergone tracheostomy 3 months prior to this admission. Pembrolizumab 100 mg (2 mg/kg) had been given every 3 weeks and she had received a radiation dose of 75 Gy in 25 fractions over the neck tumor. The cuff of a tracheostomy tube was over-inflated to temporarily stop bleeding in an emergency. In suspicion of a tracheo-innominate artery fistula, she received computed tomography angiography after initial stabilization (Figure 1 and Figure 2).

Besides adequate initial resuscitation, over-inflating the cuff of a tracheostomy tube may be an effective technique to temporarily control bleeding from tracheo-innominate artery fistulae [1,2,3,4,5]. This case demonstrated the reliability of computed tomography angiography as an imaging modality for confirmation.

## Figures and Tables

**Figure 1 diagnostics-14-00223-f001:**
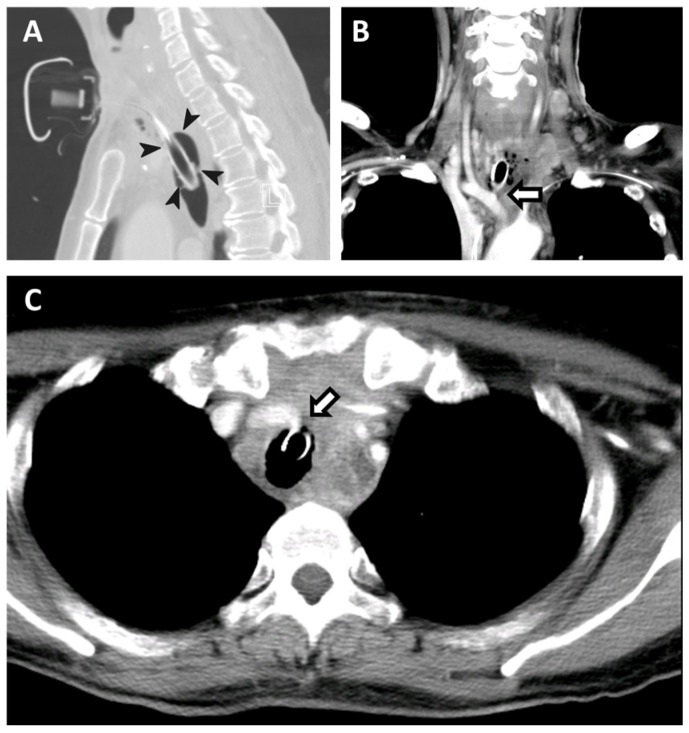
Computed tomography angiography of a 58-year-old woman with excruciating tracheostoma bleeding temporarily controlled by over-inflating her tracheostomy. (**A**) Sagittal view, (**B**) coronal view, (**C**) axial view. The images revealed contrast extravasation at innominate artery middle part (pointed by bordered box arrows) that was directly tamponaded by over-inflated tracheostomy tube cuff (demonstrated by bulging tracheal wall and circled by arrowheads).

**Figure 2 diagnostics-14-00223-f002:**
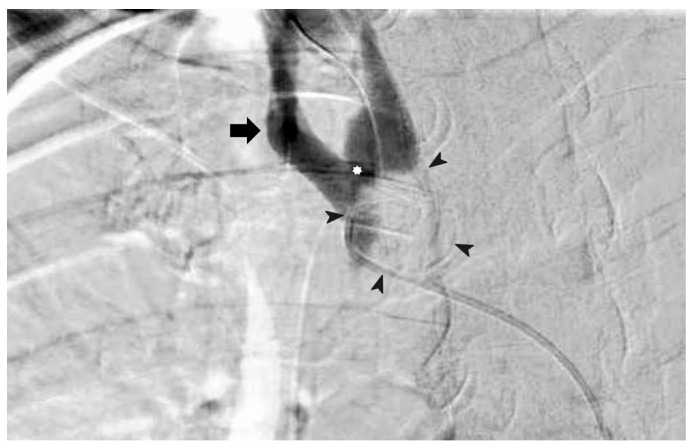
Fistula identification using angiography of the patient with a slightly deflated tracheostomy tube cuff. Flow of contrast into upper trachea through fistula (white star) at tracheostomy tube cuff (circled by arrowheads); right innominate artery (arrow). The patient was managed with endovascular stenting followed by ligation of innominate artery in a hybrid operation theater (Appendix A). Two weeks later, she was discharged to a chronic care center. The patient died peacefully at the chronic care center two weeks after discharge due to terminal disease.

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
