# Peer review of "Over-Inflating a Tracheostomy Tube Cuff for Tracheo-Innominate Artery Fistula"

_diagnostics, 2024, doi:10.3390/diagnostics14020223_

Round 1
Reviewer 1 Report
Comments and Suggestions for Authors
Dear Authors and Dear Editor
I read this interesting case/image report of a successfully bleeding control by over-inflating a tracheostomy cuff in a tracheo-innominate artery fistula reported by Yi Chia Hsieh, Wen Ruei Tang and Ying Yuan Chen.
Since this is a case report, it is not possible to raise any deep questions or doubts because in the picture legends are summarized all the relevant aspect of this case.
I would like to highlight that, unfortunately, tracheo-innominate artery fistula is more common than we think especially in this subset of patients (anaplastic thyroid carcinoma who need a tracheostomy), but I found really interesting the figure where fistula is identificated by using angiography.
It could be really interesting a picture of a CT or angiography after stenting and ligation of the innominate artery.
Author Response
Your suggestion is greatly appreciated. Therefore, we provide both an angiography stenting photo and a intraoperative picture in the supplementary. Figure S1 is attached below.

Reviewer 2 Report
Comments and Suggestions for Authors
Thank you for the opportunity to review.
Tracheo-innominate artery fistula is not uncommon.
If the imagaes indifcates a bleeding site, it may be rare.
However, there are several concerns with this report.
1. I suspect that the image appears that the cannula is in direct contact rather than the cuff. If this figure shows the cuff, what is the message to the readers? Is it just a valuable moment?
2. Even if temporary hemostasis is achieved by increasing cuff pressure, can this case be life-saving? Generally, in cases where the artery and trachea intersect, resuscitation is typically challenging.
The message to the reader is not clear. To avoid conveying incorrect messages to the readers, it is desirable to proofread the text.
Author Response
We really appreciate your comments.
1. For better visualization of arterial fistula, we set the window-level 50 HU and window-width 266 HU for evaluating the mediastinum, which made tracheostomy cuff barelly seen. Therefore, we set the window-level -500 and window-width 2000 for better cuff demonstration in the attached Figure S2.
2. We do not think that solely increasing cuff pressure is enough. As we mentioned in the abstract, both adequate initial resuscitation and over-inflated tracheostomy cuff are of utmost importance. The final paragraph has been revised to better convey our message.
*** Besides adequate initial resuscitation, over-inflating the cuff of tracheostomy is an effective technique to temporarily control bleeding from tracheo-innominate artery fistula and an invaluable bridging maneuver to surgery. Furthermore, computed tomography angiography is a reliable imaging modality to confirm diagnosis, which is clearly demonstrated in the case.

Round 2
Reviewer 2 Report
Comments and Suggestions for Authors
1. Figure 1A is unnecessary; please replace it with Figure S2B.
2. Cuff of tracheostomy is incorrect. Cuff of a tracheostomy tube is correct. Please check and modify including the title.
3. The text is too conclusive. I recommend modifying the manuscript as follows (line 37-41).
Besides adequate initial resuscitation, over-inflating the cuff of a tracheostomy tube may be an effective technique to temporarily control bleeding from tracheo-innominate artery fistula [1-5]. This case demonstrated that the reliability of computed tomography angiography as an imaging modality for confirmation.
Comments on the Quality of English Language1. Figure 1A is unnecessary; please replace it with Figure S2B.
2. Cuff of tracheostomy is incorrect. Cuff of a tracheostomy tube is correct. Please check and modify including the title.
3. The text is too conclusive. I recommend modifying the manuscript as follows (line 37-41).
Besides adequate initial resuscitation, over-inflating the cuff of a tracheostomy tube may be an effective technique to temporarily control bleeding from tracheo-innominate artery fistula [1-5]. This case demonstrated that the reliability of computed tomography angiography as an imaging modality for confirmation.
Author Response
Thank you for constructive comments. The figure and the manuscript have been modified.

Round 3
Reviewer 2 Report
Comments and Suggestions for Authors
No more comments